# Learning to Discover and Detect Objects

**Vladimir Fomenko**[1]* **Ismail Elezi**[2] **Deva Ramanan**[3]
**Laura Leal-Taixé**[2] **Aljoša Ošep**[2,3]
[1]Microsoft Azure AI [2]Technical University of Munich
[3]Carnegie Mellon University

vladfomenko@microsoft.com
{ismail.elezi, leal.taixe, aljosa.osep}@tum.de deva@cs.cmu.edu

## Abstract

We tackle the problem of *novel class discovery and localization (NCDL)*. In this setting, we assume a source dataset with supervision for only some object classes. Instances of other classes need to be discovered, classified, and localized automatically based on visual similarity without any human supervision. To tackle NCDL, we propose a two-stage object detection network *Region-based NCDL (RNCDL)* that uses a region proposal network to localize regions of interest (RoIs). We then train our network to learn to classify each RoI, either as one of the *known* classes, seen in the source dataset, or one of the *novel* classes, with a long-tail distribution constraint on the class assignments, reflecting the natural frequency of classes in the real world. By training our detection network with this objective in an end-to-end manner, it learns to classify all region proposals for a large variety of classes, including those not part of the labeled object class vocabulary. Our experiments conducted using COCO and LVIS datasets reveal that our method is significantly more effective than multi-stage pipelines that rely on traditional clustering algorithms. Furthermore, we demonstrate the generality of our approach by applying our method to a large-scale Visual Genome dataset, where our network successfully learns to detect various semantic classes without direct supervision. Our code is available at `https://github.com/vlfom/RNCDL`.

## 1 Introduction

We tackle novel class discovery and localization in unlabeled datasets, a long-standing problem in computer vision [55, 59, 41, 42, 54]. As we are approaching the limits of well-understood and successful supervised training of object detectors [22, 21, 53, 27] and labeling novel and rare classes that appear in the long tail of object class distribution is becoming prohibitively expensive [24], we expect joint novel class discovery and localization to become increasingly more important.

Object class discovery in image collections is a difficult problem, as there are, in general, many possible equally valid attributes (*e.g.*, object color or orientation) for grouping of visual patterns. As shown in [29, 30], *novel class discovery (NCD)* can be well-posed and tackled in a data-driven manner by injecting prior knowledge on how we wish to group semantic classes using some degree of supervision. However, existing NCD methods [29, 30, 25, 26, 72, 7, 73, 74, 19] all assume curated datasets, where objects of interest are pre-cropped and semantic classes are fairly balanced. By contrast, in this paper, we tackle joint *novel class discovery and localization (NCDL)*, a task of learning to discover and detect objects from raw, unlabeled, and uncurated data. This is a significantly more challenging and realistic problem setting, as images always contain a mixture of labeled and unlabeled object classes, and we need to localize and categorize them.

---

*Work done while at Technical University of Munich.

36th Conference on Neural Information Processing Systems (NeurIPS 2022).

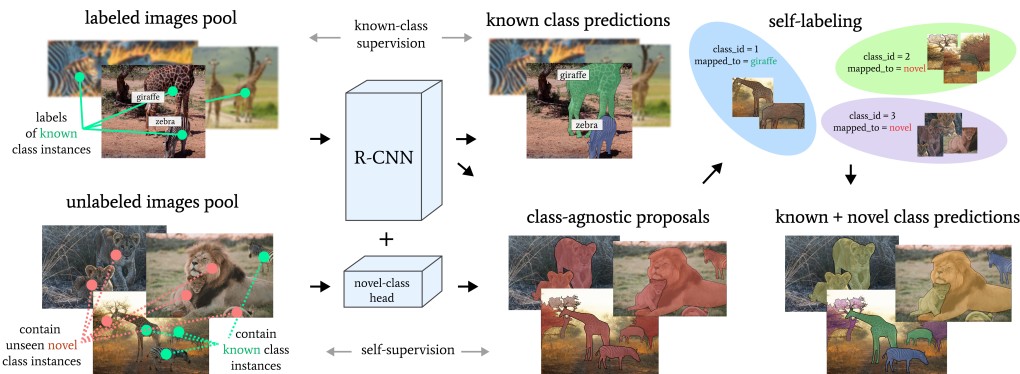

Figure 1: **Novel class discovery and localization.** Given labeled images along with annotations for *known*, frequently observed semantic classes and unlabeled images that may contain instances of *novel* classes, our network learns to localize and recognize common semantic classes and categorize novel classes, for which supervision in the form of labeled instances is not available.

We re-purpose COCO [43] and LVIS [24] datasets to study this problem. We use $50\%$ of COCO images together with COCO class vocabulary and labels for supervised model training. We treat the remaining $50\%$ of images as unlabelled image collection. We use these images to learn to detect remaining object classes for which supervision is not given. With this setting, we mimic real-world scenarios where no supervision is given for the target domain.

We train our two-stage [53, 27] object detector *Region-based NCDL* (*RNCDL*) in an end-to-end manner such that we can (i) correctly detect (*i.e.*, localize and categorize) labeled objects, and, in parallel, (ii) detect unlabeled objects by learning feature representations suitable for categorization of objects that would otherwise be classified as the background class. We start with supervised training on the labeled source domain, followed by self-supervised training on the target (unlabeled) domain. Finally, we transfer knowledge from the source to the target domain by retaining weights for class-agnostic modules, such as RPN, bounding box regression, and instance segmentation heads. For classification, we add to the primary *known* classification head a new, secondary head for categorization of regions that do not correspond to these a-priori *known* semantic classes. We train both heads jointly with an objective to categorize every region proposal consistently under multiple transformations and a constraint that the marginal distribution of class assignments should follow a long-tailed distribution. Such a non-uniform classification prior incentivizes the network to learn features that capture the diversity of objects and prevents the network from being biased towards the labeled, *known* semantic classes.

Our end-to-end trainable *RNCDL* network categorizes all regions of interest in a single network forward pass and does not require running clustering algorithms during the inference. Moreover, it significantly outperforms prior art [68] that learns object representations using contrastive learning followed by k-means clustering. More importantly, with this work, we lay the foundations for an exciting new line of research beyond well-established supervised learning for object detection.

In summary, this paper makes the following **contributions**: (i) we study the problem of *novel class discovery and localization* in-the-wild using standard object detection datasets that do not pre-localize regions of interest manually or assume objects are roughly centered, *i.e.*, exhibit a strong photographer bias. (ii) we propose a simple yet effective and scalable method that can be trained end-to-end on the unlabeled domain. First, our network is bootstrapped using a labeled dataset to learn features suitable for categorizing future instances. Then, we learn feature representations suitable for categorization for arbitrary objects using the self-supervised objective. Finally, (iii), we show that our method can discover and detect novel class instances and outperform prior work, demonstrating the generalization to datasets beyond COCO.

## 2   Related Work

**Object discovery in image and video collections.** The discovery of semantic classes that appear in the unlabeled image or video collections is a long-standing research problem [60]. Early methods

employ multiple bottom-up segmentation of images [55, 59] to discover commonly occurring patterns using topic modeling, *e.g.*, Dirichlet allocation [6].

Method by [41] relies on saliency, while [42] incorporates semantic knowledge about certain semantic classes for which supervision is available. These methods are evaluated on smaller datasets [23, 58, 17] that contain one or a handful of well-delineated objects. Rubinstein *et al.* [54] tackles object discovery methods in nosy internet photo collections. Beyond images, Kwak *et al.* [40] tackle the discovery of objects in YouTube videos that usually contain a single object exhibiting dominant motion. They build on a two-stage approach: (i) identify the most salient region proposals in terms of appearance and motion, and (ii) co-segment consistently appearing objects across video clips. Object discovery in videos, recorded from a moving platform, was tackled in [49, 50] by first mining video-object tracks by associating region proposal network (RPN) [53] based proposals across time, followed by clustering of video proposals based on features, extracted from a pre-trained network. Recent work on open-set panoptic segmentation [32] similarly groups object proposals based on features extracted from a pre-trained network to pseudo-label commonly-occurring objects. The method by [68] follows the general approach of clustering RPN proposals; however, instead of relying on pre-trained CNN features, this approach learns representations suitable for clustering using self-supervised objective, trained via contrastive learning.

While most of the foregoing works focus on saliency criteria to significantly narrow down the number of region proposals, [62, 63] resort to combinatorial optimization that can cope with a large set of object proposals to discover objects that frequently appear in image datasets. Furthermore, it has been shown [64] that this approach can be scaled to large image collections, *e.g.*, OpenImages [39].

**Novel class discovery.** *Novel class discovery* (NCD) methods assume labeled data for some semantic classes is given. This data can be used as a guide to learn a notion of similarity and invariances to certain features that should be ignored for the grouping (such as object color and orientation). Unlabeled images are assumed to contain novel class instances only. Hsu *et al.* [29, 30] propose a framework that injects prior knowledge on how we wish to group semantic classes from supervised learning on labeled images to the unlabeled domain via learnable similarity function. The work of Han *et al.* [25] builds on deep clustering [71] and suggests an approach for automatically discovering the number of novel classes. Han *et al.* [26] build on the pairwise similarity of image features [12] and propose leveraging self-supervised pretraining along with freezing the backbone to reduce bias towards the known classes. Follow-up works [72, 34, 74, 73] further boost the performance by leveraging self-supervised learning and regularization techniques. UNO [19] proposes a simple method based on SwAV [11] that bootstraps signal for novel class training using pseudo-labels generated under equipartitioning constraints. ORCA [7] extends the NCD problem setting and assumes that we are given images containing instances of *novel* and *known* objects at inference time. Motivated by [61, 12, 65], they leverage self-supervised initialization of backbone, supervised loss, and unsupervised loss based on pairwise pseudo-labels, and novel dynamic softmax margin [65] to alleviate bias of the known classes. By contrast to our work, the aforementioned NCD methods investigate the problem in the image classification setting, assuming objects of interest are localized (pre-cropped) and operate in the absence of outliers (*e.g.*, images containing no objects).

**Related and complementary research directions.** Methods for generic [45] or large vocabulary object detection [24] focus on detecting/segmenting a large set of a-priori known semantic classes, often by utilizing multiple datasets and (possibly weak) supervision [75, 52, 31]. A special case is zero-shot learning [70] and detection [3, 48, 51], where (mostly weak) supervision for unseen classes comes in the form of attributes, class names, latent features, or other forms of auxiliary data commonly used to learn a joint multi-modal embedding space. Few-shot learning methods [36, 67, 18, 8] learn from one or a few data samples per class. Methods by [18, 8] employ a secondary classification head that is trained using few-shot supervision and distillation [18] or margin loss [8]. In contrast to zero-shot, few-shot, and large-vocabulary detection, we assume no supervision and prior knowledge for object classes that appear in the tail.

Methods for open-set recognition [56, 57, 33, 5] and detection [48, 16] focus on the calibration of per-class uncertainties to minimize the confusion between known and possibly unknown classes. As defined by [4, 46], open-world recognition methods must explicitly recognize *unknown* object instances that were not observed as labeled samples during the model training and must continually update object detectors to recognize these unknown instances. Recent work on open-world detection [35] focuses on the continual learning aspect of the task. For the evaluation, labels for *novel*

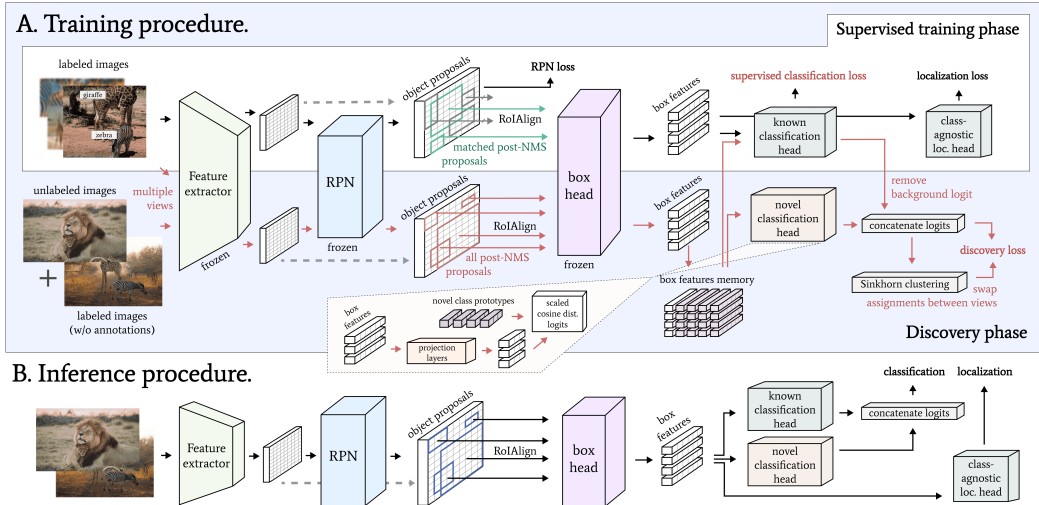

Figure 2: A high-level overview of our network. *(top)* **During the supervised training phase**, we train our backbone and RPN networks using *labeled* data, together with classification head and a class agnostic localization head. **During the discovery phase**, we freeze all the layers of the network apart from classification head and attach and train a novel classification head using *unlabeled* data. **During the inference (bottom)**, we perform a standard R-CNN pass, using classification heads of both known and novel categories to predict a class assignment for each proposal. This can be either one of $K$ classes, that were presented a labeled samples during the model training, or any novel object class that appears in the training data.

classes are fully provided (those are the held-out classes). Our work is orthogonal and tackles the "missing bit" of the open-world detection pipeline: semantic grouping of *novel* objects that can appear in the training data. Clusters of discovered novel classes could be verified by human annotators and used to update detection models continually, as proposed by [4].

## 3 Novel Class Discovery and Localization

We first detail novel class discovery and localization (NCDL), the task of learning to discover and detect objects from unlabeled and uncurated data. We assume given a source (*labeled*) image dataset $D^k$ for *known* object classes $C^k$ ($|C^k| = K$), that provides both class and box level objects supervision: $D^k = \{(I, \{(\mathbf{b}, c)\})\}$, where $I \in \mathbb{R}^{h \times w \times 3}$, $\mathbf{b} \in \mathbb{R}^4$, and $c \in C^k$. In addition, we use one or more target (*unlabeled*) datasets $D^n = \{I\}$ that may contain instances of both previously seen and *novel* object classes $C^k \cup C^n$ ($|C^n| = N, N > K$) that we need to learn to detect in the absence of any labels. For an arbitrary image $I$ from the source or target domain, NCDL methods should detect all object instances present in the image: $\{(\mathbf{b}', c')\}$, *i.e.*, localize them *and* categorize them. For example, an instance of a known class, *e.g.*, car, needs to be detected as the `car` class, while a novel class instance with $c' \in C^n$ needs to be assigned to one of the unknown, *novel* classes (*e.g.*, `unknown_1`, `unknown_2`, etc.).

As this paper's main contribution, we propose *Region-based NCDL* (*RNCDL*), an end-to-end trainable detection and discovery network that learns to localize and categorize objects, including those for which explicit supervision is not given. We base our method on a two-stage Faster or Mask R-CNN [53, 27], trained in a supervised manner on a dataset that contains labels for *some* semantic classes. Such detectors can already reliably localize and classify *known* objects and are naturally suitable for the NCDL task, as their region proposal network (RPN) is trained in a class-agnostic manner to propose regions of interest (RoIs) that likely contain objects. Given object-centric feature representation for each localized RoI, one natural approach for discovering (categorizing) novel classes would be to cluster such regions in feature space using, *e.g.*, k-means, *c.f.*, [49, 32, 68]. Feature representations learned via supervised R-CNN training are suitable for localizing regions that likely contain objects and for classifying them as one of the *known* classes. However, features

representing RoIs that do not contain labeled class objects are effectively trained to be collapsed into a single *background* class and, therefore, not discriminative enough for robust categorization of novel instances. Furthermore, such an approach would have a strong bias toward classifying novel instances as one of the labeled classes.

## 3.1 Method Overview

In the following, we detail our *RNCDL*, which learns to categorize all region proposals jointly on both (labeled) source and (unlabeled) target datasets in an end-to-end manner. In a nutshell, we first train class-agnostic modules of our detector on the source domain (Fig. 2A, top), *i.e.*, RPN, bounding box regression head, (optionally) instance segmentation head, and a classifier for *known* classes. We proceed with self-supervised training (Fig. 2A, bottom) on the target domain to learn to additionally categorize instances of *unknown* classes. To this end, we add a secondary classification head (to the primary classification head, trained to categorize *known* classes) and train both jointly, with an objective to categorize every region proposal consistently under multiple transformations and a constraint that the marginal distribution of class assignments follows a long-tailed distribution.

**Supervised training on the source domain.** We begin with a self-supervised backbone pre-training [28] on ImageNet [15] to initialize the R-CNN backbone network. We then train the object detector on the source (labeled) domain using supervision for objects of known $C^k$ classes, such as 80 COCO classes (Fig. 2A, top). We use the labeled data to train the shared feature extractor (backbone) together with a class-agnostic region proposal network (RPN), class-agnostic box regression head, and a classification head for the categorization of $C^k$ classes. Specifically, we train RPN to predict a set of class-agnostic proposals $\{(\mathbf{b}, o)\}$, for each image, where $\mathbf{b} \in \mathbb{R}^4$ are predicted coordinates and $o \in \mathbb{R}$ is the objectness score. We then use RoIAlign [27] to crop RoI features from the backbone and feed them to an MLP to obtain box-head features $\hat{\mathbf{f}} \in \mathbb{R}^{\hat{F}}$ that we pass to the additional network heads to obtain the final predictions. We train a class-agnostic bounding box regression head to refine box location for each RoI $\hat{\mathbf{b}} = h^{bb}(\mathbf{b}, \hat{\mathbf{f}})$, and a classification box head to categorize each RoI as one of the $K$ classes or the *background* class: $\hat{c} = h^k(\hat{\mathbf{f}}) = \mathbf{W}^k \hat{\mathbf{f}}$, where $\hat{c} \in \mathbb{R}^{K+1}$ and $\mathbf{W}^k \in \mathbb{R}^{(K+1) \times \hat{F}}$. We train our network using the standard R-CNN loss [53] $L_{sup} = L_{RPN} + L_{box} + L_{cls}$, that consists of RPN classification $L_{RPN}$ loss, box regression loss $L_{box}$, and second-stage classification loss $L_{cls}$. We discuss all changes to Detectron2 Faster R-CNN implementation, as needed for the NCDL task, in Sec. **??** in the appendix.

**Discovery on the target domain.** For the self-supervised (discovery) training phase on the target domain, we first modify the primary (*known*) classification head by removing the *background* class from its weight matrix such that $\hat{\mathbf{W}}^k \in \mathbb{R}^{K \times \hat{F}}$. Then, we attach a secondary (novel) classification head $h^n$ on top of RoI box features $\hat{\mathbf{f}}$ (Fig. 2A). This head consists of an MLP and a linear classification layer that we detail in Sec. 3.2.

In this phase we freeze all layers except the classification heads. We train both classification heads jointly using a self-supervised classification loss $L_{ss}$ (detailed in Sec. 3.3) and supervised classification loss $L_{cls}$: $L_{disc} = L_{ss} + \alpha \cdot L_{cls}$ where $\alpha$ is the supervised loss scale coefficient. We show in our ablations (Sec. 4.2) that keeping the downscaled ($\alpha < 1$) supervised classification loss in the loop effectively alleviates forgetting the weights for the known classes, while large $\alpha$ may induce an unwanted bias towards classifying RoIs as *known* classes.

Both classification losses are effectively cross-entropy losses. $L_{cls}$ is trained using provided labels, while $L_{ss}$ is trained using pseudo-labels generated online for the current batch of proposals. We detail online pseudo-label generation in Sec. 3.3. In a nutshell, this loss encourages the network to categorize every region proposal consistently under multiple transformations and, importantly, under a constraint that the marginal distribution of predicted class assignments for the last batches follows a prior long-tailed distribution. We compute the supervised loss only for annotation-matched region proposals. With such a joint training objective, our network is encouraged to learn to distinguish a large variety of classes present in the regions sampled by RPN while retaining the capability to classify known objects.

## 3.2 Secondary Classification Head

As the box features $\hat{\mathbf{f}}$ are kept frozen during the discovery phase, we use several non-linear projection layers in the secondary classification head: $\hat{\mathbf{f}}^n = g^n(\hat{\mathbf{f}})$, where $g^n$ is a multi-layer perceptron with

ReLU activations. With this transformation, we learn to disentangle features for distinct *novel* classes without affecting the RoI box features $\hat{\mathbf{f}}$. On top of the projected features $\hat{\mathbf{f}}^n$, we employ a cosine classification layer, shown to be effective in the context of few-shot learning [20, 13, 66]: $\mathbf{l}^n = \frac{\mathbf{W}^n}{||\mathbf{W}^n||} \cdot \frac{\hat{\mathbf{f}}^n}{||\hat{\mathbf{f}}^n||}$, where $\mathbf{l}^n \in \mathbb{R}^N$ are the predicted novel-class logits. We determine the number of novel classes (the parameter $N$ for the secondary head) empirically (see Sec. 4.2).

### 3.3 Self-Supervision via Online Constrained Clustering

For self-supervised learning, we perform pseudo-labeling via constrained clustering during online model training to assign a soft pseudo-label to each RoI (RPN proposal). We generate pseudo-labels $\mathbf{q} \in \mathbb{R}^{(N+K) \times B}$ for RoIs in all input images (labeled and unlabeled, $D^N \cup D^K$) that can span all classes, both *known* and *novel*. The parameter $B$ denotes the batch size. We use these pseudo-labels $\mathbf{q}$ as a supervisory signal to compute cross-entropy loss $L_{ss}$ to train both classification heads.

For pseudo-label generation one possibility would be to use standard clustering methods (*e.g.*, k-means [47]), however, such offline clustering would make training slow, and such methods were shown to produce unstable results in domain of image classification [9, 10]. Instead, we follow recent developments in the field of self-supervised representation learning [11, 2, 1] and minimize a clustering energy function online during the mode training (*i.e.*, cross-entropy loss): $E(\hat{\mathbf{p}}, \mathbf{q}) = \frac{1}{B} \sum_{i=1}^{B} \sum_{y}^{N+K} q(y|I_i) \log \hat{p}(y|I_i)$, where $\mathbf{q}$ are class labels (*i.e.*, pseudo-labels), and $\hat{\mathbf{p}} = \text{softmax}([\mathbf{l}^k, \mathbf{l}^n]) \in \mathbb{R}^{(N+K) \times B}$ are predicted class-probabilities. Note that both labels $\mathbf{q}$ and probabilities $\hat{\mathbf{p}}$ are subject to optimization. Similar to [2], we employ an alternating algorithm. First, we use a constrained clustering method to generate pseudo-labels $\mathbf{q}$ based on current network weights. Then, we update weights by optimizing $\hat{\mathbf{p}}$ using fixed $\mathbf{q}$, and iterate.

To generate class pseudo-labels, we first choose a target clusters' marginal probability distribution (*i.e.*, a prior distribution). Empirically, we obtained best results using a non-symmetrical log-normal distribution. This distribution is right-skewed, suitable for modeling the long-tail. We then generate such soft pseudo-labels $\mathbf{q}$ that (i) closely match the chosen marginals and (ii) minimize $E(\hat{\mathbf{p}}, \mathbf{q})$ for the given $\hat{\mathbf{p}}$. In [2, 1], it is shown that this can be posed as a constrained optimization problem and solved as an optimal transport problem using the fast online Sinkhorn-Knopp algorithm [14]. Having obtained the pseudo-labels $\mathbf{q}$, we fix them, compute the loss $L_{ss} = E(\hat{\mathbf{p}}, \mathbf{q})$, and update weights for the classification heads using the back-propagation algorithm.

**Memory module.** To generate good-quality pseudo-labels for each mini-batch, we need a diverse set of samples that capture a wide variety of classes. A small set of features per mini-batch could lead to noisy cluster assignments. As we operate in scenarios with more than a thousand classes to discover, we introduce a memory module (*c.f.*, [28, 11]) to store RoI box features from the last batches and ensure a diverse set of samples when computing clustering assignments. As in [28, 11], we design the memory module as a queue, where at the end of each iteration, we replace the oldest set of stored features with the features from the current batch. During pseudo-labeling, we pass the extended set of features to the Sinkhorn algorithm. Specifically, after extracting current-batch RoI features, we concatenate them with the features stored in the memory module $\mathbf{M}$, compute their logits, and as an input to the Sinkhorn algorithm use: $\hat{\mathbf{l}} = [\hat{\mathbf{l}}^k, \hat{\mathbf{l}}^n] = [h^k([\hat{\mathbf{f}}, \mathbf{M}]), h^n([\hat{\mathbf{f}}, \mathbf{M}])]$, where $\mathbf{M} \in \mathbb{R}^{M_{sz} \times \hat{F}}$ and $M_{sz}$ is the number of features stored in the memory. Such a modified procedure generates $B + M_{sz}$ pseudo-labels for both current-batch proposals and memory features. We discard pseudo-labels generated for memory features and use only those $B$ generated for the current-batch proposals, *i.e.*, we use the memory module only to improve the quality of current-batch pseudo-labels.

**Swapping multi-view cluster assignments.** To ensure that the network outputs consistent classification predictions $\hat{\mathbf{p}}$, we employ an additional supervisory signal from multiple augmentations [11]. Specifically, for each image from $D^N \cup D^K$, we first generate RoIs (proposals) and refine their coordinates via a class-agnostic classification head. We then generate two augmented views for each image (we discuss augmentations in Sec. **??** in the appendix), and extract features twice for each RoI. Finally, we generate pseudo-labels for each view (set of features) individually, using the Sinkhorn algorithm as described in Sec. 3.3, obtaining two sets of pseudo-labels $\mathbf{q}_i$, $i \in \{1, 2\}$ that correspond to the same proposals. To enforce feature invariance to the augmentations and capture the semantic similarity of the proposals, we swap the resulting pseudo-labels $\mathbf{q}_i$ between views and calculate losses as $E(\hat{\mathbf{p}}_i, \mathbf{q}_j)$, where $i \neq j$, *i.e.*, we use pseudo-labels obtained for one view, as labels to the other. We

calculate the total loss as the average of the losses for both views: $\hat{L}_{ss} = (E(\hat{\mathbf{p}}_1, \mathbf{q}_2) + E(\hat{\mathbf{p}}_2, \mathbf{q}_1))/2$. For each view, we maintain a separate feature memory module.

### 3.4 Inference and Evaluation

Finally, to categorize RoIs (RPN proposals) during the inference (Fig. 2B), we extract a set of RoIs $\{(\mathbf{b}, o)\}$ for each image and pass them to localization refinement and class prediction heads. We pass RoI features through both $h^k$ and $h^n$ classification heads and concatenate their logits: $\hat{\mathbf{p}} = \mathrm{softmax}([\mathbf{l}^k, \mathbf{l}^n]) = \mathrm{softmax}([h^k(\hat{\mathbf{f}}), h^n(\hat{\mathbf{f}})]) \in \mathbb{R}^{K+N}$ to compute output class probabilities over $K$ *known* and $N$ *novel* semantic classes.

**Semantic class assignment.** Our method provides only categorization for *novel* classes. Class IDs (*i.e.*, cluster IDs) are not directly mapped to classes, defined in a certain semantic class vocabulary, as needed for the evaluation. We therefore need to obtain a mapping of predicted class IDs to ground-truth semantic clsses. To do so, we follow similar strategy as prior art [25, 26, 19, 68]. Once we train our network, we generate classification predictions for each GT annotation in the validation dataset. Then, we use the Hungarian algorithm [38] to obtain an one-to-one mapping between GT labels and validation class predictions. We ignore instances of predicted classes that were not mapped to GT classes. Then, we proceed to the standard R-CNN inference pass, where we apply the generated mapping and follow the standard R-CNN post-processing and evaluation [43].

## 4 Experimental Evaluation

In this section, we discuss our evaluation test-bed, including datasets, evaluation settings, and metrics (Sec. 4.1). In Sec. 4.2 we justify our main design decisions by studying NCDL performance in a well-controlled setting that closely follows a real-world scenario (training on a source *labeled* and target *unlabeled* dataset). We then compare our method's performance to several baselines (Sec. 4.3) and, finally, demonstrate that our method is applicable beyond our evaluation test-bed (Sec. 4.4).

### 4.1 Evaluation Setting

**COCO$_{half}$ + LVIS.** We re-purpose COCO 2017 [43] and LVIS v1 [24] datasets for running ablations and comparisons with baselines and prior art. We use annotations for 80 COCO classes during the supervised training phase, aiming at further classifying additional 1000+ LVIS classes. COCO dataset contains $123K$ images with modal bounding box and segmentation mask annotations for 80 classes. The LVIS dataset contains a subset of $120K$ COCO images with annotations for 1203 classes that include all classes from COCO. We follow LVIS training and validation splits for our experiments, resulting in $100K$ training images and $20K$ validation images. In our NCDL setup, we further split $100K$ training images in half. We use only $50K$ images with 80-class annotations from COCO during the supervised training phase. We treat the rest of $50K$ images as additional unlabeled data used during self-supervised training (the discovery phase).

**LVIS + Visual Genome.** We perform a large-scale generalization experiment by using annotations for 1203 LVIS classes during the supervised learning (*i.e.*, treating LVIS classes as *labeled*), aiming to learn to discover extra 2700+ classes from the Visual Genome (VG) v1.4 dataset [37] (*i.e.*, treating VG classes as *unlabeled*). We provide dataset details and splits (that ensure LVIS and VG class vocabularies do not overlap) in the supplementary. We note that annotations provided in VG are not exhaustive per class, and many of its classes are abstract or semantically overlapping both within VG and when compared to LVIS. We thus mainly focus on qualitative results.

**Implementation details.** We base our model on Mask R-CNN [27] and FPN [44] implementations from Detectron2 [69]. We discuss implementation details in Sec. **??** of the appendix.

**Evaluation metrics.** For quantitative evaluation, we follow [25, 26, 19, 68] and match predicted cluster assignments with annotated semantic classes (Sec. 3.4). We follow common practice and report mean average precision (mAP@[.5:.95]) [43]. For qualitative results, we include instances of classes that were matched to annotated classes only. We detail our evaluation procedure in the supplementary.

Table 1: Impact of supervised loss strength. We check the results of our method as a function of the strength of the supervised loss.

| Sup. loss coeff. | mAP$_{all}$ | mAP$_{known}$ | mAP$_{novel}$ |
|---|---|---|---|
| 0 | 6.59 | 16.58 | 5.77 |
| 0.1 | 6.90 | 22.26 | 5.64 |
| **0.5** | **6.92** | **25.00** | **5.42** |
| 1.0 | 6.35 | 25.81 | 4.75 |

Table 2: Sensitivity to the number of classes. We vary the number of novel classes set during the discovery phase.

| # novel classes | mAP$_{all}$ | mAP$_{known}$ | mAP$_{novel}$ |
|---|---|---|---|
| 1000 | 5.42 | 23.24 | 3.95 |
| **3000** | **6.92** | **25.00** | **5.42** |
| 5000 | 6.24 | 24.91 | 4.70 |

**Baselines.** We compare our end-to-end trainable detector with prior work by Weng *et al.* [68], NCD methods ORCA [7], UNO[19], and k-means [47] baseline that all operate on cropped object proposals, provided by RPN. For NCD methods, we use labeled instance annotations from COCO$_{half}$ dataset as labeled images pool and cropped RPN proposals as unlabeled images pool. We extract proposals using the same R-CNN base network trained on COCO$_{half}$ to ensure comparable evaluation conditions. After training NCD classifiers, we use the same R-CNN to generate object proposals for the validation images, apply the classifiers on top of the proposals, and continue with the standard R-CNN post-processing steps. We provide more details in Sec. **??** in the appendix.

## 4.2 Model Ablations

We ablate single and multi-phase training, the impact of keeping supervised loss in the loop, the number of novel categories, pseudo-labeling prior, model architecture decisions, and backbone pretraining. In Sec. **??** in the appendix, we provide further results on experiments with hyperparameters for proposals extractor and how class-agnostic heads affect the performance of the supervised baseline.

**Single-phase training and unfreezing R-CNN components during discovery.** As shown in Table 3, we observe that training the network with both supervised and discovery losses from the beginning, summed up without any scaling, leads to divergence. We hypothesize that this is due to noisy gradients from our self-supervised signal, with R-CNN being very sensitive to the choice of losses and hyperparameters [53]. To further support this claim, we experiment with a standard supervised phase followed by a discovery phase without freezing any components. We observe that such a setup also leads to divergence. Unfreezing only the RoI heads during the discovery phase does not lead to divergence, however, it reduces the score by $1.32$ mAP. We thus keep all the components frozen during the discovery phase, apart from known and novel classification heads.

**The strength of the supervised loss.** We ablate the strength of the supervised classification loss that is used along with the discovery loss during the discovery phase. We vary the strength from $0$ (no supervised loss) to $1$ (no downscaling) and provide the results in Table 1. We observe that excluding supervised loss completely degrades the network's performance for the known classes and benefits the performance of the novel classes, while keeping the full supervised loss introduces a bias towards the known classes. This indicates a tradeoff between performance on known and novel classes. We set the strength of the supervised loss to $0.5$ in all experiments.

**Sensitivity to the number of novel classes.** In Table 2 we experiment with varying the number of novel classes we use for the novel classification head. We observe that the best number of classes for the COCO + LVIS setup is 3000, while other choices result in lower scores. We conjecture that such scores are sensitive to the semantic classes defined by the target dataset – with 3000 classes, the model learns the granularity and grouping that matches closest to the LVIS classes, but that does not necessarily mean that a finer or coarser granularity categorization is incorrect. In Sec. **??** in the appendix, we also experiment with attaching and training multiple heads jointly during the discovery phase and observe that scores slightly degrade in such setup but do not observe a significant drop in results.

**Clustering prior.** In Table 3, we experiment with using uniform prior distribution for class-marginals during pseudo-labels generation, as proposed in [2, 10]. We observe degradation of the score by $1.17$ mAP, which confirms our hypothesis that a non-uniform log-normal prior distribution better models a real-world long-tailed class distribution. We use k-means for the online pseudo-labels generation as an additional ablation instead of constrained clustering. As a result, we observe a significant score degradation by $4.86$ mAP, especially notable for the novel classes.

Table 3: Additional ablations. We experiment with removing individual components from the network and their impact on the overall performance.

| Method | mAP$_{all}$ | mAP$_{known}$ | mAP$_{novel}$ |
|---|---|---|---|
| **RNCDL** | **6.92** | **25.00** | **5.42** |
| Single-phase framework | n/a | n/a | n/a |
| Unfreeze all layers | n/a | n/a | n/a |
| Unfreeze RoI heads | 5.60 | 20.55 | 4.37 |
| Uniform class prior | 5.75 | 24.11 | 4.23 |
| k-means [47] pseudo-lab. | 2.06 | 21.13 | 0.48 |
| W/o projection MLP | 5.30 | 26.13 | 3.58 |
| W/o swapped assignments | 6.74 | 26.02 | 5.14 |
| W/o memory | 2.83 | 24.19 | 1.06 |
| W/o MoCo pretraining | 4.77 | 16.92 | 3.77 |

Table 4: Comparison with state-of-the-art models. * as per open source code. † adapted to support known classes in the target dataset. ‡ randomly initialized novel head.

| Method | mAP$_{all}$ | mAP$_{known}$ | mAP$_{novel}$ |
|---|---|---|---|
| *Methods that operate on cropped proposals* | | | |
| k-means [47] | 1.33 | 15.61 | 0.14 |
| Weng *et al.* [68]* | 1.62 | 17.85 | 0.27 |
| ORCA [7] | 2.03 | 20.57 | 0.49 |
| UNO [19]‡ | 2.18 | 21.09 | 0.61 |
| *Methods that operate on FPN-based features* | | | |
| k-means [47] | 1.55 | 17.77 | 0.20 |
| RNCDL w/ random init.‡ | 1.95 | 23.51 | 0.17 |
| **RNCDL** | **6.92** | **25.00** | **5.42** |
| Fully-supervised | 18.47 | 39.38 | 16.74 |

**Embedding projector and swapped assignments.** In Table 3, we demonstrate the effect of adding an MLP feature projector on top of RoI box features for novel head classification. We observe that the performance of the network improves by $1.62$ mAP. Furthermore, the same table shows that swapping pseudo-labels between proposals under multiple augmentations helps boost the score by $0.18$ mAP.

**Feature memory module.** We demonstrate the importance of the memory module in Table 3. In all our experiments, we used the memory size of 100 batches unless stated otherwise. Removing the memory module significantly drops the performance by $4.09$ mAP, especially for the novel classes.

**Self-supervised pretraining matters.** We compare the results of our framework under different backbone initialization employed for the supervised pretraining stage. For the randomly initialized backbone, we set all layers as trainable, while during the MoCo initialization, the first two convolutional blocks are kept frozen (*c.f.*, [28]). In Table 3 we observe that the MoCo-initialized backbone improves the score by $2.15$ mAP over randomly initialized backbone mode. We only pre-train our network using ImageNet dataset [15], and not on the source or target object detection dataset.

**Proposals extractor hyperparameters.** We observe that when extracting the RPN proposals for self-supervised learning during the discovery phase, the absence of NMS yields the best results, and the optimal number of proposals per image is 50. We thus use such parameters for all the experiments.

## 4.3 Comparison to Prior Art

In Table 4, we compare our method with k-means baselines and three recent state-of-the-art methods that we adapted to our scenario. We present the results for both *known* (COCO) and *unknown* (the rest of LVIS) classes. Our method significantly outperforms the previous best NCD method UNO [19] by reaching $4.74$ higher overall mAP. Furthermore, we outperform UNO in both the *known* and especially *novel* classes, where we reach $4.81$ higher mAP. We also significantly outperform the approach by Weng *et al.* [68]. Our method is the only method to reach over $1$ mAP in novel classes, where we reach $5.42$ mAP. We believe the main advantages of our method to be the memory module, critical for effective self-supervision, and a backbone trained in an end-to-end manner with the aid of class-agnostic losses, frozen during the discovery phase. Further, our backbone benefits from high-level semantic features via FPN that are critical for the classification of small objects [44]. We provide more comparison details in the supplementary. We note that as our method heavily relies on the generated region proposals, we are limited by the performance of the region proposal network (*i.e.*, we cannot detect objects not detected by the RPN network). We do not update the backbone feature extractor during self-supervised training. Successfully doing so could further boost the performance, including the localization and segmentation components.

## 4.4 Cross-dataset Generalization and Qualitative Results

We analyze the generalization properties of our framework by training on LVIS and evaluating on Visual Genome (VG) dataset. We present quantitative results in Table 5.

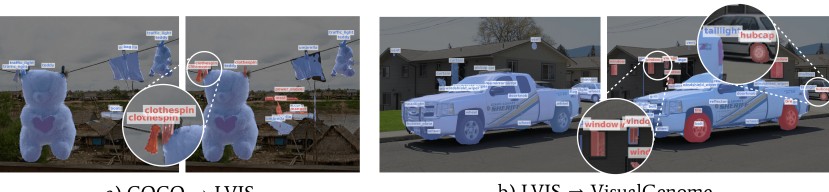

a) COCO → LVIS                    b) LVIS → VisualGenome

Figure 3: Visualization of predictions for validation images of the fully-supervised model and our RNCDL framework. We color the discovered novel classes in red.

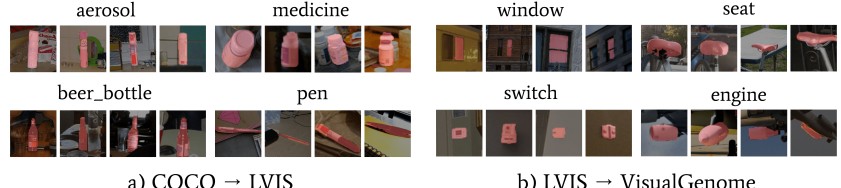

a) COCO → LVIS                    b) LVIS → VisualGenome

Figure 4: Visualization of the common novel classes discovered in LVIS and VisualGenome datasets.

In Figure 3, we provide qualitative results with novel classes discovered. In Figure 4, we also visualize the commonly discovered categories. Upon manual examination of the clusters from the LVIS + VG experiment, we observe that many novel classes are synonyms, hyponyms, or hypernyms of known classes appearing in the LVIS dataset.

Table 5: LVIS → VisualGenome comparisons with a fully-supervised method.

| Method | mAP$_{all}$ | mAP$_{known}$ | mAP$_{novel}$ |
|---|---|---|---|
| RNCDL w/ random init. | 2.13 | 7.71 | 0.81 |
| RNCDL | 4.46 | 12.55 | 2.56 |
| Fully-supervised | 4.52 | 13.72 | 2.35 |

## 5   Conclusions

In this paper, we introduce an end-to-end RNCDL network for novel class discovery, detection, and localization tasks. Our model is a two-stage object detection network that can classify both instances of the labeled, known classes and those of unlabeled, novel classes. At the core of our method is a self-supervision guided by features of region proposals and a constraint for the class assignments to follow a long-tail distribution. In our experiments, we demonstrate a significant improvement over the previous state-of-the-art. Furthermore, we demonstrate the ability to detect semantic classes without any supervision at a large scale on the Visual Genome dataset.

## Broader Impact

The real world contains a vast number of object categories, but labeling them all is impractical and costly. Most deep learning models for object detection are trained on datasets that cover only a fixed and limited set of categories, and they cannot generalize to novel classes that are not seen during training. This limits their applicability in scenarios where objects of interest are unknown or rare. Therefore, there is a need for methods that can discover and detect novel classes without relying on labeled data supervision. This work presents a novel end-to-end method that addresses this problem and significantly improves over previous approaches. We hope our work will inspire more research on this challenging problem and lead to more general and robust object detection models.

**Acknowledgments.** LLT would like to acknowledge the Humboldt Foundation through the Sofja Kovalevskaja Award. DR would like to acknowledge the CMU Argo AI Center for Autonomous Vehicle Research. We thank the anonymous reviewers for their feedback.

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
