# A  Abstract

In this supplementary, we provide implementation details (Section B), R-CNN configuration details (Section C), details on LVIS and Visual Genome datasets (Section D), results for additional ablations (Section E), details on reproducing the prior works (Section F), segmentation and scores of higher granularity for the best-scoring models (Section G), and additional qualitative examples (Section H).

# B  Implementation Details

Our model is based on Mask R-CNN [11] configuration from Detectron2 [23] with ResNet50-based [10] FPN [16] backbone. Different from the standard configuration, we use AMP [19], SyncBatchNorm [24], MoCo [12] for backbone initialization, and class-agnostic localization and mask heads. In section C, we provide more details and analysis of the introduced modifications.

During the supervised training phase, for all experiments, we follow Detectron2 and use a batch size of 16, train all networks using SGD optimizer for $180K$ iterations, and use random resize and random flip augmentations. We linearly increase the learning rate from $10^{-3}$ to $10^{-2}$ for the first $1K$ iterations and decrease it tenfold at iterations $120K$ and $160K$. When switching between $COCO_{half}$ + LVIS and LVIS + VG setups, architecture-wise, we only change the number of target classes in the known classification head.

During the discovery training phase, we train for $15K$ iterations with a learning rate of $10^{-2}$ and follow a cosine decay schedule with linear ramp-up. The learning rate is linearly increased from $10^{-5}$ to $10^{-2}$ for the first $3K$ epochs and then decayed with a cosine decay schedule [17] to $10^{-3}$.

For the supervised forward pass during the discovery training phase, we do not change the hyperparameters of the R-CNN and use the exact same model configuration with the same batch size, base learning rate, and set of input images (re-shuffled). However, we remove the background (*unmatched*) RoI proposals from supervised pass training and keep only proposals that got matched to annotations. We also use a supervised loss scale coefficient of $0.5$, which results in an effective learning rate for the supervised loss to be twice smaller than that of the discovery loss.

For the discovery forward pass, we use a separate pool of unlabeled images (that include images from the labeled pool). For RPN hyperparameters, we extract 50 proposals per image and do not use NMS. We use a memory module with a size of $20K = 100 \cdot 4 \cdot 50$ features per GPU, where 100 is its size (in the number of batches stored), 4 is the per-GPU batch size, and 50 is the upper limit of the number of RPN proposals extracted per image in the batch. As the memory is empty at the beginning of training, we start training only after 150 iterations (batches) to allow it to get filled with features. We use a number of iterations slightly higher than the memory size to account for images with less than 50 proposals. For self-supervision prior, similarly to [1], we let marginals follow $\text{Lognormal}(1, 0.5) \cdot \frac{M}{N}$, where $M$ is the number of samples used for Sinkhorn and $N$ is the total number of classes. In our experiments, $M$ is calculated dynamically, based on the number of proposals in the current batch, and, at most, equals $20.2K$ (where $20K$ and $200$ are the memory size and the maximum number of proposals in the batch respectively). For $N$ we use 3080 classes in the $COCO_{half}$ + LVIS setup and $4203 = 1203 + 3000$ classes in the LVIS + VG setup. We set $\lambda = 20$ for the Sinkhorn algorithm. To obtain multi-view features for each proposal, we use augmentations similar to the ones from SimCLR [3]. Specifically, we use weaker versions of random color distortion (color jitter, grayscale), random Gaussian blur, and random resizing. We do not use random cropping to avoid cropping out some proposals only for one of the views. We use a batch size of 16 that is split across 4 GPUs, resulting in a per-GPU batch size of 4. We use $torch.distributed$ backend [14] for training, with the memory module and pseudo-labels being local to each of the 4 GPU processes.

For evaluation on LVIS and VG datasets, we follow LVIS evaluation specification [9] and use up to 300 top-scoring predictions with a per-prediction confidence cutoff of $1e^{-4}$. We use 4x NVIDIA A40 GPUs for all experiments.

# C  R-CNN Configuration Details

In Table C we provide details on modifications introduced to the Detectron2's default configuration that we used for R-CNN models. We provide scores for the fully-supervised models trained on

COCO$_{half}$ and LVIS datasets. The models marked in bold were used for initialization during the discovery phases in COCO$_{half}$ + LVIS and LVIS + VG setups, respectively. To the default configuration, we first introduced AMP [19] to allow for training larger models during the discovery phase and SyncBatchNorm [24] to stabilize training. Then, we replaced supervised ImageNet [6] initialization for the backbone with self-supervised initialization. We avoided using supervised initialization to ensure that no supervision was given for the novel classes we aim to discover, as some of the ImageNet classes overlap semantically with classes from LVIS and VG. Specifically, we use MoCo v2 [5] applied on ImageNet dataset [6] and use weights from a public repository provided by the authors of the paper from `https://github.com/facebookresearch/moco`. Finally, we switch to class-agnostic localization & mask heads.

Table C1: **Performance of fully-supervised models on COCO$_{half}$ and LVIS datasets**. In bold, we highlight the configuration used for the supervised training phase and the fully-supervised baseline.

| Model | COCO$_{half}$ mAP | LVIS mAP |
|---|---|---|
| Default Detectron Mask R-CNN FPN Res50 | 34.92 | 18.87 |
| + add AMP, SyncBN | 34.07 | 18.03 |
| + replace ImageNet w/ MoCo init | 35.78 | 18.85 |
| **+ use class-agnostic heads** | **35.69** | **18.47** |

# D    LVIS + Visual Genome Setup

For the LVIS + Visual Genome (VG) setup, we use annotations from the LVIS v1 [9] dataset and aim to discover novel classes present in the Visual Genome (VG) v1.4 dataset [13]. The LVIS dataset contains $120K$ images with annotations for 1203 classes. Its training and validation splits contain $100K$ and $20K$ images, respectively. VG dataset contains $108K$ images with annotations for more than $7K$ categories mapped to WordNet [20] synsets. $50K$ of VG images overlap with LVIS dataset images. To simplify the training and evaluation of experiments, we define the validation split of the VG dataset as a set of images from the LVIS validation split that also appear in VG. This results in $100K$ and $8K$ images for VG training and validation splits, respectively. For the supervised training phase, we use $100K$ images from the LVIS train split with LVIS annotations. During the discovery phase, for the unlabeled pool, we merge LVIS and VG training images, resulting in $158K$ images total. We use $8K$ images from the VG validation split for evaluation. Out of more than $7K$ synsets from VG, we keep only those that appear in both generated training and validation VG splits, resulting in 3367 classes in total. We observe that they include 641 LVIS classes by using exact matching of Wordnet synsets of classes, leaving 2726 novel classes to discover. We note, however, that with such exact mapping, we do not account for potential overlapping word senses. Moreover, annotations provided in VG are not exhaustive per class, and many of its classes are abstract. As a result, many categories of the resulting dataset may be semantically overlapping, and we note that quantitative mAP results provided may not reflect the *real* model's performance.

# E    Additional RNCDL Ablations

Table E1: **Ablations for RPN hyperparameters, memory size, and multi-head setup.** We study the results of our method as a function of a) the NMS coefficient used by RPN and RoI heads to generate class-agnostic proposals during the discovery phase, b) the number of RPN proposals, c) memory size in the number of mini-batches stored. In d), we demonstrate that it's possible to train the model in a multi-head setup with a varying number of classes without substantial loss in performance. We train the model with multiple heads of 1000, 3000, and 5000 classes and then evaluate each of the heads separately.

| NMS coeff. | mAP$_{all}$ |
|---|---|
| 0.8 | 5.76 |
| 0.9 | 6.77 |
| 0.99 | 6.84 |
| **no NMS** | **6.92** |

(a)

| # proposals | mAP$_{all}$ |
|---|---|
| 20 | 6.49 |
| **50** | **6.92** |
| 100 | 6.86 |
| 150 | 6.75 |

(b)

| # mem. batches | mAP$_{all}$ |
|---|---|
| 0 | 2.83 |
| 20 | 4.66 |
| 50 | 5.83 |
| **100** | **6.92** |

(c)

| # novel classes | mAP$_{all}$ | mAP$_{known}$ | mAP$_{novel}$ |
|---|---|---|---|
| 1000 | 5.37 | 25.87 | 3.68 |
| 3000 | 6.34 | 25.56 | 4.76 |
| 5000 | 5.14 | 25.25 | 3.48 |

(d)

**NMS coefficient for proposals.** In Table E1a, we ablate the strength NMS used by RPN to filter out class-agnostic overlapping proposals during the discovery phase. Note that this only concerns the coefficient used for filtering the proposals generated for the discovery phase training and not the NMS used during inference. We observe that a larger (weaker) NMS yields better results, and the absence of NMS yields the best results. This demonstrates that the network benefits from observing more proposals per image during training, even if they highly overlap.

**Number of proposals per image.** We experiment with extracting a different number of RPN proposals per image during the discovery phase in Table E1b. We observe that the optimal number of proposals is 50, and having more proposals per image damages its performance.

**Memory module size.** We experiment with different memory module sizes, expressed in the number of last mini-batches stored, in Table E1c. We observe a large jump in performance of $1.83$ mAP when introducing a memory module of 20 batches. Afterward, the performance tends to grow as its size increases roughly log-linearly. Due to memory and time limitations, we only experiment with memory sizes at most 100 and expect the models' performance to improve further with more features stored.

**Sensitivity to the number of novel classes for a multi-head setup.** In Table E1d, we experiment with attaching and training multiple heads jointly during the discovery phase, where the total discovery loss is computed as an average loss per head, and the memory module is shared across heads. We observe that scores degrade in such a setup but do not observe a substantial drop in results. This suggests that for a practitioner, it is possible to train a single framework with multiple heads, each with hypothesized number of categories to discover and then to use any of the heads for the desired grouping.

**FC instead of cosine classification layer.** We experiment with replacing the cosine classification layer [8, 4, 21] of the novel classification head with an FC layer and observe that the training diverges.

# F  Details on Baselines and Prior Work Reproduction

In this section, we provide implementation details for baselines and prior works that we compare. For all the methods, we use $3000$ for the number of novel categories.

For comparison with Weng *et al.* [22] work, we trained their method in our setting using the code provided by authors available at `https://github.com/ZZWENG/longtail_segmentation`. As the method starts with a similar supervised training phase, for a fair comparison, we used an R-CNN trained in the same manner as in RNCDL experiments. We followed the authors' configuration and extracted the top $50$ proposals per image with an NMS of $0.75$ before the clustering phase. *For the clustering phase we found multiple inconsistencies with the code provided by the authors and the original manuscript. Even after correspondence with the authors, we could not resolve all issues and reproduce results comparable to those reported in [22]. We thus report results that best reflect the foregoing method according to the available official implementation.*

For comparison with NCD works, which operate on images tightly cropped to the object of interest, we used localization coordinates of annotations and generated proposals to obtain a set of image crops. Specifically, for *labeled* images pool, we used annotations from $COCO_{half}$ dataset, and for *unlabeled* images pool, we used predicted RPN proposals. We used these annotations and proposals to crop images to the box coordinates of each instance. This resulted in more than $350K$ labeled crops and $5M$ unlabeled crops. We then padded each crop to a squared shape using the mean pixel value of all crops and scaled them to the size of $224 \times 224$ to match ImageNet images specification. To extract RPN proposals, we trained an R-CNN network on $COCO_{half}$ with the same configuration as during the supervised training phase to ensure comparable evaluation conditions. We then applied state-of-the-art ORCA [2], UNO [7], and baseline k-means [18] method to the resulting labeled and unlabeled crops to train image classifiers. Having trained the classifiers, we followed a standard R-CNN inference loop, replacing the default FC classification head with each of the resulting classifiers. We do not alter other R-CNN procedures, such as post-processing, NMS, etc.

ORCA [2] is directly applicable to such an image classification setup. We followed the configuration used by the authors for the ImageNet-100 dataset, but used a batch size of 1024 and trained for 10

Table G1: Detailed results for the RCNDL model.

| Classes | mAP | mAP$_{50}$ | mAP$_{75}$ | mAP$_s$ | mAP$_m$ | mAP$_l$ | mAP$_r$ | mAP$_c$ | mAP$_f$ |
|---------|-----|-----------|-----------|---------|---------|---------|---------|---------|---------|
| known | 25.00 | 42.41 | 24.93 | 13.25 | 32.41 | 34.56 | - | - | 25.00 |
| novel | 5.42 | 8.00 | 5.81 | 2.62 | 6.05 | 9.40 | 8,02 | 5.75 | 3.55 |
| all | 6.92 | 10.63 | 7.27 | 3.57 | 8.31 | 12.22 | 8.02 | 5.75 | 7.74 |

(a) Detection scores for the RNCDL model in COCO$_{half}$ + LVIS setup

| Classes | mAP | mAP$_{50}$ | mAP$_{75}$ | mAP$_s$ | mAP$_m$ | mAP$_l$ | mAP$_r$ | mAP$_c$ | mAP$_f$ |
|---------|-----|-----------|-----------|---------|---------|---------|---------|---------|---------|
| known | 25.21 | 40.69 | 26.12 | 11.23 | 31.77 | 39.98 | - | - | 25.21 |
| novel | 5.16 | 7.44 | 5.48 | 2.09 | 6.19 | 9.51 | 8.03 | 5.29 | 3.39 |
| all | 6.69 | 9.97 | 7.06 | 2.91 | 8.39 | 12.92 | 8.03 | 5.29 | 7.65 |

(b) Segmentation scores for the RNCDL model in COCO$_{half}$ + LVIS setup

| Classes | mAP | mAP$_{50}$ | mAP$_{75}$ | mAP$_s$ | mAP$_m$ | mAP$_l$ | mAP$_r$ | mAP$_c$ | mAP$_f$ |
|---------|-----|-----------|-----------|---------|---------|---------|---------|---------|---------|
| known | 12.55 | 21.43 | 12.48 | 6.24 | 14.59 | 18.25 | 13.33 | 10.24 | 13.91 |
| novel | 2.56 | 3.95 | 2.65 | 1.39 | 2.04 | 3.15 | 3.02 | 2.83 | 1.72 |
| all | 4.46 | 7.28 | 4.52 | 2.47 | 4.79 | 6.22 | 3.34 | 4.05 | 5.69 |

(c) Detection scores for the RNCDL model in LVIS + VG setup

epochs with a learning rate annealed by a factor of 10 at epochs 6 and 8. UNO [7] operates under the assumption that images from the unlabeled pool may only contain novel classes. We thus extended the method to allow known classes to be present in the unlabeled pool. We implemented this by allowing pseudo-labels generated at each iteration to contain known classes. We followed the configuration used by the authors for the ImageNet dataset but used a batch size of 1024, and trained the supervised learning phase for 100 epochs and the discovery learning phase for 20 epochs. For k-means [18], we used a contrastive self-supervised method [12] to initialize a backbone and fine-tune it with images from the labeled pool. We then extracted features for all the images and learned cluster centroids. During the evaluation, as k-means does not output confidence, we classified each RPN proposal based on the closest cluster centroid and marked its confidence as 100%. In another k-means ablation, we applied the same methodology on top of RoI box head features of instances and proposals without training a new backbone.

Finally, we also experimented with evaluating the RNCDL model right after attaching the novel head, with weights for MLP projection and cosine classification layers initialized randomly.

## G  Segmentation, Per-Area, and Per-Frequency Results

In Table G1 we provide additional quantitative results for the best-scoring RNCDL models from COCO$_{half}$ + LVIS and LVIS + VG setups. Specifically, we report mAP for `small`, `medium`, and `large` object instances, following COCO [15], and `rare`, `common`, and `frequent` object classes, as proposed in LVIS [9]. In addition, we report segmentation scores of our Mask R-CNN model for COCO$_{half}$ + LVIS setup. We do not report segmentation scores for LVIS + VG setup as VG annotations do not contain segmentation masks.

## H  Additional Qualitative Examples

We provide more qualitative results for both COCO$_{half}$ + LVIS and LVIS + VG setups. For both setups, we use larger models and train longer to obtain clusters of higher quality. Specifically, we train for $25K$ iterations, use a learning rate of $0.015$, and use 8 GPUs with a batch size of 4 per GPU, making the effective batch size 32.

In Figures H1 and H2 we provide more qualitative examples for COCO$_{half}$ + LVIS and LVIS + VG setups, respectively. In addition, we highlight novel classes, determined according to the mapping procedure, in red.

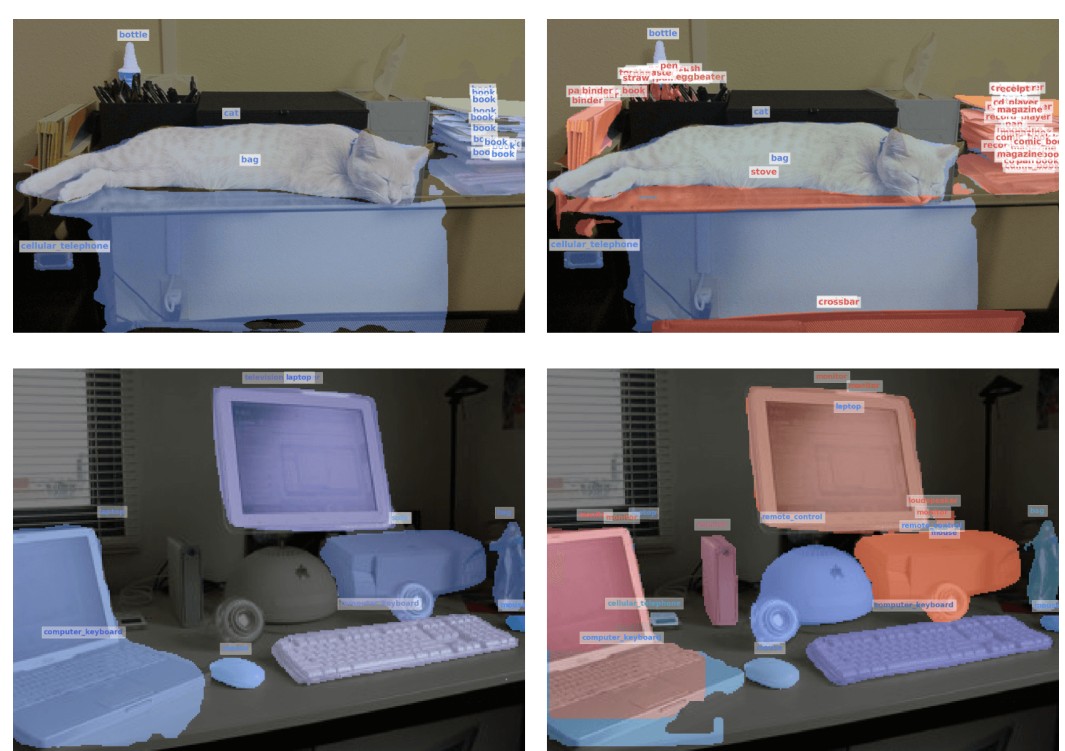

a) COCO → LVIS

Figure H1: Visualization of predictions for validation images of the fully-supervised model and our RNCDL framework in COCO$_{half}$ + LVIS setup. We color the discovered novel classes in red.

In Figure H3 we provide the top clusters ordered by confidence and size that were not mapped to one of the *known* classes by the mapping procedure. For the most confident clusters, we observe that many of them semantically overlap with the known annotated classes. This could indicate that the network splits some known classes into sub-clusters. For most of the largest clusters, we observe that their content can be characterized as noisy, and most images do not contain a clear object instance present. This could indicate that the network learns to group noisy proposals.

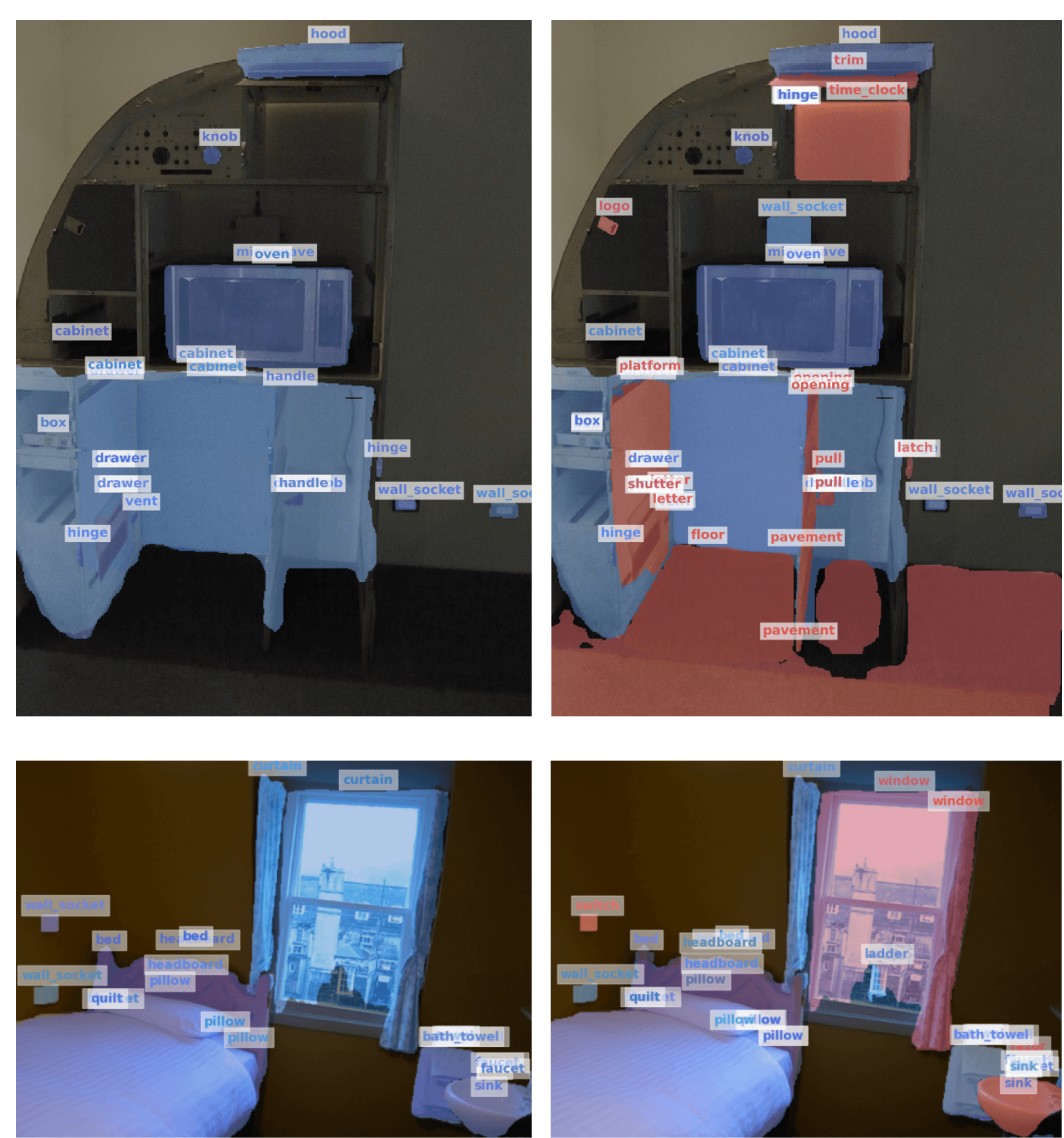

b) LVIS → VisualGenome

Figure H2: Visualization of predictions for validation images of the fully-supervised model and our RNCDL framework in LVIS + VG setup. We color the discovered novel classes in red.

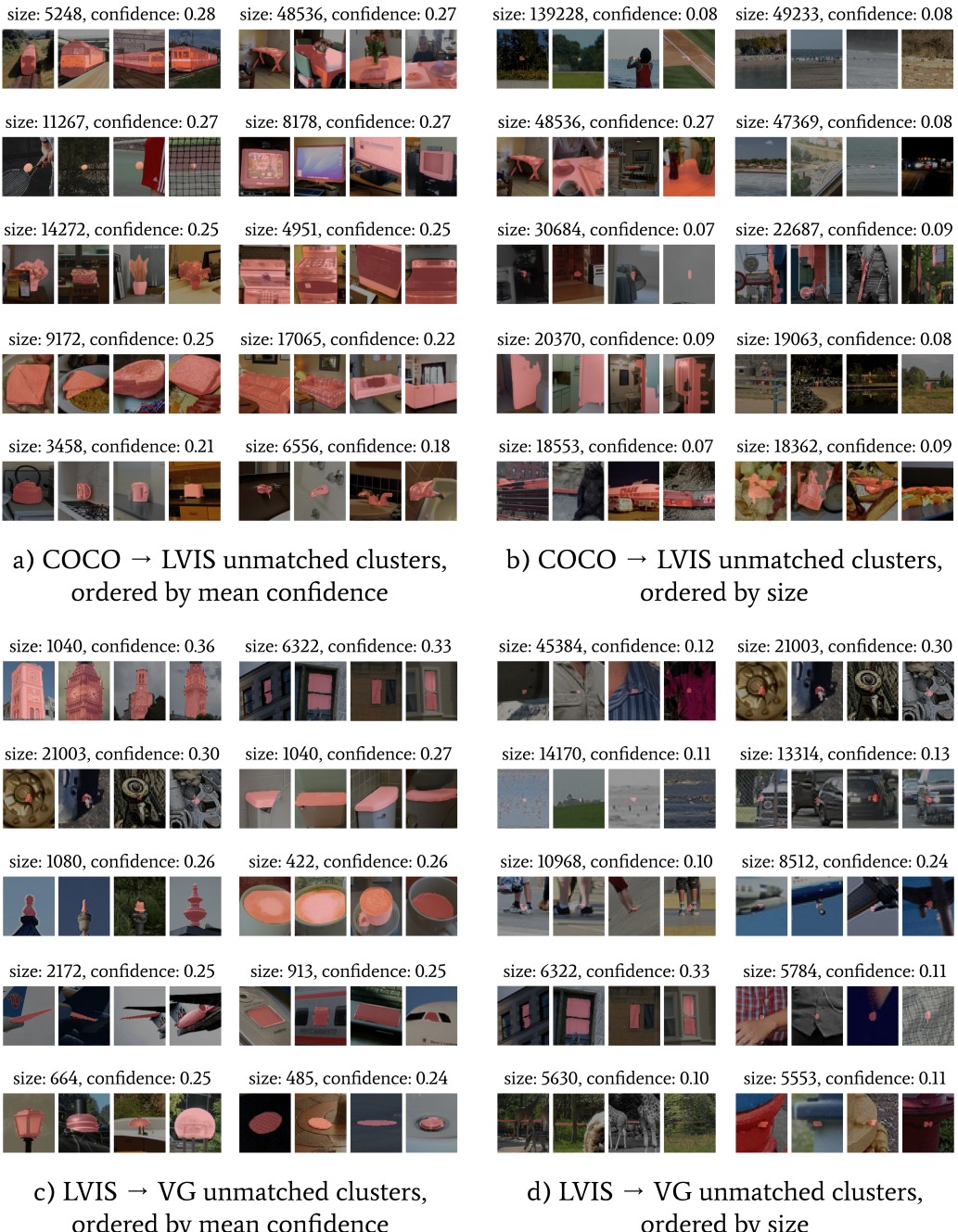

a) COCO → LVIS unmatched clusters, ordered by mean confidence

b) COCO → LVIS unmatched clusters, ordered by size

c) LVIS → VG unmatched clusters, ordered by mean confidence

d) LVIS → VG unmatched clusters, ordered by size

Figure H3: The largest and the most confident clusters discovered.