# OpenReview forum: "Learning to Discover and Detect Objects"
_NeurIPS.cc/2022/Conference — NeurIPS 2022 Accept_

### Official Review · Reviewer_mWgP · 2022-07-09

**Rating:** 4
**Confidence:** 4
**Soundness:** 3 good
**Presentation:** 3 good
**Contribution:** 2 fair

**Summary:**

This paper investigates the problem of novel class discovery, detection, and localization, which trains a model with a labeled source dataset and unlabeled target datasets that can discover, classify and localize novel classes. This paper proposed a two-stage object detection network and outperforms traditional clustering algorithms and self-supervised contrastive learning methods.

**Questions:**

See weaknesses.

**Limitations:**

Yes.

**Strengths And Weaknesses:**

**## Strengths ##**

1. The setting investigated in this paper is interesting and practical.
2. The proposed method achieves good performance on complex datasets.
3. Extensive ablation studies demonstrate the effectiveness of each component.

**## Weaknesses ##**

1. The novelty of the method is limited since the techniques introduced in this paper have been proposed by other NCD and clustering methods. For example, self-supervised discovery [17], second classification head [17], and swapping cluster assignments [1,2]. This paper only replaces the image features with RoI features in the detection task.

2. In Sec 4.2, this paper investigates the number of novel classes, COCO+LVIS setting only contains about 1200 classes, while 3000 novel classes are used. How to assign the 3000 classes to specific classes?

3. This paper leverages [1,2] for generating pseudo-labels. Kmeans is only compared as the baseline. What if using kmeans to generate pseudo-labels while still using other techniques?

4. In the supplement, the memory size is only 100 batches per GPU, but there exist 3000 novel classes. Is such a small memory enough?

5. What does the fully-supervised method in Tab. 5 denote? Is it trained on LVIS or VisualGanome? It is kind of unconvincing if the fully-supervised method is trained on VisualGanome.

---

> ### Author Response · Authors · 2022-08-02
> **Response to Reviewer mWgP**
>
> We are grateful to the reviewer for carefully reading our paper and for overall positive feedback. We are happy that the reviewer finds our investigation setting **interesting and practical**, the proposed method to be **well-performant** and our ablation studies **extensive**, demonstrating the **effectiveness** of each component. In the following, we address the reviewer's comments.
>
> **The reviewer suggests that the novelty of our paper is limited, as several components have been proposed in the context of self-supervised representation learning (swapping cluster assignments) and in the context of image recognition. The reviewer also comments that “this paper only replaces the image features with RoI features in the detection task".**
> We would like to point out that it is difficult to respond to the comment on limited novelty, as to the best of our belief, this is subjective; according to B97E our paper is “one of the pioneering works in discovering novel objects by self-supervised technology in object detection field”. To properly respond, it would be helpful to know whether limited novelty is meant in terms of methodology, problem formulation, or potential paper impact. The statement that the paper “only replaces the image features with RoI features in the detection task” is not correct. Indeed, we build on the RoI features from the object detector, however, our setup introduces several notable challenges not present in the prior works that we detail below.
>
> NCD methods [7,19,25,26,29,30,34,72,73,74] assume only novel-category objects in the target domain. Different from them, self-labeling via clustering [1, 2] and SwAV [11], we take on the more challenging problem of large-scale object detection. The foregoing works operate on artificially balanced pre-cropped image patches with no outliers. In contrast, we tackle the problem of learning object-centric, clusterable representations from open-ended uncurated image collections.
>
> Our work is the first to explore the application of lognormal prior distribution as a clustering constraint on top of RoI object features in combination with the memory module. Finally, we show the emerging capability of our method to group and discriminate noisy proposals, along with discovering both rare, common, and frequent classes.
>
> **The reviewer asks about how we assign the 3000 predicted novel classes to COCO+LVIS classes for the evaluation.**
> To match the predicted clusters with annotated classes we employ the Hungarian algorithm on top of predictions for localized ground truth object instances. Having obtained the 1:1 mapping, we then discard the ground truth annotations. Finally, we perform inference to generate object proposals, classify them, and use the foregoing mapping to obtain classes that match the target dataset's classes and can be used for evaluation. We now moved this explanation to Sec. 3 for completeness.
>
> **The reviewer asks for an additional experiment, where K-means is used to generate pseudo-labels.**
> As per reviewer’s suggestion, we added such pseudo-labeling ablation to Tab. 3 in the paper. Using k-means for pseudo-labeling we obtain a result of 2.06 mAP (compared to 6.92 mAP obtained by our proposed method).
>
> **The Reviewer points out that the memory size is only 100 batches per GPU, while there are 3000 novel classes, asking if the memory size is large enough.**
> In the supplementary, we describe the 100 batches stored in the memory corresponding to 20K proposals' features. We also highlight that such memory is unique per each GPU (x4) and per each augmented view (x2). Thus, there are 8 replicas of such memory modules, and 8x different pseudo-labels generated at each iteration. The Sinkhorn clustering is applied to each of the modules (and batches) in isolation. Such replication helps to fight the noise in the training signal, as the final loss is averaged across views and GPUs. Such a framework results in a good empirical performance. To address the reviewers' mWgP and u1PQ concerns, we added memory size ablation to the supplementary section E and Table E1c. We demonstrate that the memory size is critical to the method's performance, which improves as the memory size grows. We have revised the main text and supplementary, along with providing the ablation.
>
> **The reviewer asks for clarification regarding fully-supervised entry (trained on VisualGenome) in Tab. 5 and raises a concern about its low performance.**
> The main reason behind our method's close performance to the supervised counterpart is the nature of the VisualGenome dataset.
> The VisualGenome dataset contains many classes that semantically overlap (e.g., synonyms, hypernyms, hoponyms) [31,37]. As our framework is designed for single-label multiclass classification, its performance on such a setup, which highly resembles multi-label formulation, is expected to be low. Moreover, some classes in VisualGenome are rather abstract and describe "stuff" rather than "thing" (e.g., floor, sky).

---

> > ### Comment · Reviewer_mWgP · 2022-08-07
> > **Thanks for the rebuttal**
> >
> > Thanks for the rebuttal and most of my concerns are addressed.
> >
> > However, I admit that the setting is new and interesting, but I still do not think that self-supervised discovery, second classification head, and swapping cluster assignments can be claimed as contributions. I keep my rating but accepting it would not be that bad.

---

> > > ### Author Response · Authors · 2022-08-09
> > > **Thanks for the response**
> > >
> > > We thank the reviewer for carefully reading our response. We are thrilled that our rebuttal clarified the reviewer's concerns. We respect the reviewer's point of view and the final rating.
> > >
> > > Below, we provide further clarifications and our point of view for the potential reviewer/AC discussions.
> > >
> > > **Reviewer's view on the novelty of this paper:**
> > > > Self-supervised discovery, second classification head, and swapping cluster assignments can not be claimed as contributions.
> > >
> > > These are not claimed as contributions of this paper. A short clarification is below.
> > >
> > > **This paper's contributions**
> > >
> > > This work tackles novel class discovery *and* localization (NCDL) in the context of object detection (as explained in the paper and rebuttal). *Different* from works on Novel Class Discovery (NCD), we are not given information on which images (RoIs in our setup) contain "known" and "novel" objects — in fact, many proposals will inherently be outliers, a significant challenge not studied in the context of image classification/NCD, where datasets are curated. For the same reason, the object class distribution will be long-tailed (by contrast to e.g., ImageNet). How to cope with these challenges? All these questions are answered and distilled via our network design:
> > >
> > > * *Two-head K+N classification:* instead of supervised K+1 classification, we perform K+N classification. We classify *every* proposal using a loss that combines cross-entropy supervised loss and self-supervised loss based on swapping cluster assignment [11] objectives, followed by a concatenation of logits for the final class assignment. This network design combined with joint labeled/self-supervised training was crucial to prevent the strong bias towards classifying novel instances as one of the labeled classes. Equally important, we retain the capability to classify known instances correctly.
> > > * *Long-tailed, open-ended distribution:* Object detection is rather different from recognition. We have no control over the balance of semantic classes that appear in images. We impose lognormal prior distribution as a clustering constraint to cope with inherently long-tailed distribution.
> > > * *Scaling:* Our method was designed for a large-scale class discovery setup, aiming to discover 1000+ categories, including rare and fine-grained ones, a problem significantly different from the majority of NCD works and the references mentioned by the reviewer. One key component that enables us to do this is the memory module we use to store past RoI features during the network training.
> > >
> > > All aforementioned design decisions are experimentally justified, as recognized by all reviewers. Beyond technical contribution  (i.e., our end-to-end trainable network), we also propose a test-bed for apple-to-apple comparison of future efforts in this field in conjunction with several baselines, and demonstrate cross-dataset generalization of our method.
> > >
> > > **Final remarks**
> > >
> > > We do not believe that the prior work on self-supervised representation learning should be excluded from the design-space of self-supervised object detection and do not assume the reviewer has meant this — if so, however, it is unclear how to progress in the field. We believe that the existing literature should be a source of inspiration and not a defining factor for rejection, especially in works that open new potential research directions.

---

### Official Review · Reviewer_B97E · 2022-07-09

**Rating:** 8
**Confidence:** 4
**Soundness:** 3 good
**Presentation:** 2 fair
**Contribution:** 4 excellent

**Summary:**

This paper proposes a well-designed object detection pipeline to discovery novel objects given known objects annotations. The key ingredient is training the classification branch in roi-head with a combination of supervised loss and self-supervised loss, where the self-supervised loss is to force the object features into a set of clusters and these cluster ids could be used as the novel object category id. The experiment are well-organized, the quantitative and visual results are pretty promising.

**Questions:**

As the statement in weakness, this paper misses some important details:
- In Line 9-10, "distribution of class assignments should follow natural long-tail distributions". In Line 191, "current-batch cluster assignments to follow Gaussian distribution" and in Line of 35 of supplementary material, the marginals distribution is set as N(1, 0.5)·M/N. Why is a Gaussian distribution equal to a long-tail distribution? I find a similar implementation in reference [1], and read [1] for many times, but still can't understand. Could the author provide some more clear explanations?
- In Line 195, "cluster assignments are then used to calculate cross-entropy loss". Is the cluster assignment the hard version (0/1) or soft version as in SwAV[10]?  Is the cross-entropy loss a softmax version or a sigmoid version?
- In Line 230, "a standard forward RCNN pass based on concatenated logits from both classification heads". Does the logits include background class? Since the background logit is removed and no background proposals are trained during the discovery phase, how does the network know which proposal is background in inference?
- Although the method section "self-supervision via swapping cluster assignments" is based on reference[1,10], the authors are supposed to use some formulas to set up the problem, as method section of reference[1,10].
- In Line 256, "we follow [...] to match predicted cluster assignments with annotated semantic classes". Is this a Hungarian matching? I suggest the authors to provide more details about how to match here, since [...] reference works are mostly about image classification, and this paper is about object detection.
-  By the way, do the authors have a plan to release the code? Novel object discovery is a very important problem in object detection, releasing the code will be a good contribution to the community,

I think this paper is an obviously-accept paper, but my initial score to this paper is 6. If the authors could clarify these details in rebuttal, I will increase my score to 7. Further, if the authors could further re-write the section of method and make it self-contained, I will increase my score to 8.

References:
- [1] Yuki M. Asano, Mandela Patrick, Christian Rupprecht, and Andrea Vedaldi. Labelling unlabelled videos from scratch with multi-modal self-supervision. In NeurIPS, 2020.
- [10] Mathilde Caron, Ishan Misra, Julien Mairal, Priya Goyal, Piotr Bojanowski, and Armand Joulin. Unsupervised learning of visual features by contrasting cluster assignments. NeurIPS, 2020.

**Limitations:**

The authors didn't discuss the limitations of their work. However, it is ok for me, since this paper is one of the pioneering works in discovering novel object by self-supervised technology in object detection field.

This paper studies on object detection problem, which is a neutral computer vision technology and has no specific negative societal impact.

**Strengths And Weaknesses:**

Strengths:
- As far as I know, this work is one of the first works to study novel object detection by self-supervised technology and a well-qualified pipeline is set up accordingly.
- The novel object discovery is modeled as a clustering problem and solved by Sinkhorn Clustering algorithm. This is a good application of self-supervision on object detection.
- The experiment settings are very convincing, including cross-category(COCO-LVIS) and cross-dataset(LVIS-Visual Genome) generalization. The visualization looks amazing.

Weaknesses:
- The biggest weakness of this paper is its writing quality in the section of method. The authors didn't explain important details (see Questions below) and rely on many reference works. This is definitely wrong, because a good paper should be self-contained.
- In Line 165, "we freeze all the layers apart from the primary (known) classification head". Does it mean the whole novel object discovery phase is transformed to a simple linear problem given a set of fixed object features?

---

> ### Author Response · Authors · 2022-08-02
> **Response to Reviewer B97E**
>
> We are grateful for the highly-positive review and thrilled that the reviewer finds our detection method **well-designed, experiments organized and convincing, appreciates our cross-category and cross-dataset experiments** and finds our paper to be an **obviously-accept paper**. In the following, we address the remaining concerns.
>
> **The reviewer suggests that the biggest weakness of our paper is the presentation of the method section, which is not self-contained and omits several (important) details. The reviewer asks for a revision and provides several good hints on which parts lack clarity.**
> We thank the reviewer for this comment. As suggested, we extensively revised Sec. 3, following the suggestions of reviewers B97E and yVTK. Please find revised parts marked with blue color.
>
> **In Line 165, "we freeze all the layers apart from the primary (known) classification head".
> Reviewer asks if freezing all layers (apart from primary classif. head) implies that the whole novel object discovery phase is transformed to a simple linear problem given a set of fixed object features.**
> In the discovery phase we indeed freeze most of the R-CNN layers apart from primary and secondary (novel) classification heads. The secondary classification head's architecture includes a two-layer MLP (in addition to a cosine classification layer) that enables the head to modify (project) the RoI features for better disentanglement of features for objects of novel classes. In Table 3 (row 6) we demonstrate that such MLP significantly boosts the method's performance, especially on the novel classes. In the revised main text, we have clarified this.
>
> **The reviewer asks for a clarification on why we mention employing the Gaussian distribution in our paper.**
> The reviewer is correct that our statements were inaccurate. In our implementation, we use a Lognormal(1, 0.5) · M / N distribution as our prior, as it more closely resembles long-tailed distributions (compared to prior work that relies on uniform [1] or Gaussian [2] prior). We revised the relevant statements accordingly.
>
> **The reviewer asks if the hard or soft version (c.f., SwAV) of cluster assignments is used, and whether the cross-entropy loss is a softmax version or a sigmoid version.**
> We generate soft cluster assignments and use the standard cross-entropy loss that operates on softmax probabilities. We clarify this in the revised manuscript.
>
> **In Line 230, "a standard forward RCNN pass based on concatenated logits from both classification heads".
> Standard forward pass concatenates logits from both classification heads. Reviewer asks for clarification whether logits include background class (since background logit is removed).**
> During the discovery phase we completely remove the background logit (class) from the primary classification head. We are motivated by the fact that the secondary classification head can now learn to discriminate between novel categories, which were previously (during the supervised training phase) collapsed into the background class. We observe that during the discovery phase the network learns to filter and group remaining noisy proposals into clusters, as we demonstrate in supplementary H and Figure 3 (b, d). We clarify this in the revised manuscript.
>
> **The reviewer asks for clarification on how the predicted cluster assignments are matched with annotated semantic classes.**
> To match the predicted clusters with annotated classes we employ the Hungarian algorithm on top of predictions for localized ground truth object instances. We then discard the ground truth annotations, while keeping the resulting mapping. Finally, we perform inference to generate object proposals, classify them, and use the foregoing mapping to obtain classes that match the target dataset's classes and can be used for evaluation. Originally we explained the matching process in the supplementary. However, to address the reviewers' B97E and yVTK concerns we moved this section to the main text.
>
> **The reviewer asks whether we are planning to release the code.**
> We do plan to release the code and experimental data. We have uploaded (anonymized) code for our method, experiments and evaluation at https://github.com/5da15f23f9/learning-to-discover-and-detect-objects, for the camera-ready we are planning to clean and re-organize the code.

---

> > ### Comment · Reviewer_B97E · 2022-08-03
> > **Response to rebuttal**
> >
> > The rebuttal well-answers my questions. And the authors provide the code by the anonymized link, which I really appreciate. I decide to increase my score to 8.
> >
> > By the way, I don't think it's trivial to transfer the proven ideas in the image-level field to the detection task. This is exactly the contribution of this work. Not every image-level idea can be easily applied to roi-level and successfully benefit the object detection task.

---

> > > ### Author Response · Authors · 2022-08-05
> > > **Thank you**
> > >
> > > We thank the reviewer for increasing their initial score, and we are happy that our rebuttal has well-answered the reviewer's questions.
> > >
> > > Furthermore, we echo the reviewer's comment that it is far from trivial to just transfer ideas from the image-level field to object detection, which comes with a massive amount of challenges not present in the image-level field.

---

### Official Review · Reviewer_u1PQ · 2022-07-11

**Rating:** 5
**Confidence:** 3
**Soundness:** 3 good
**Presentation:** 3 good
**Contribution:** 2 fair

**Summary:**

This paper proposes the RNCDL network for novel class discovery, detection, and localization (NCDL). The model is firstly supervised and trained on the source domain for known classes and then self-supervised trained on the target domain to discover unknown classes. During discovering, the self-supervision is guided by the feature of region proposals and a constraint for the novel class assignments to follow a long-tail distribution. Extensive experiments on COCO, LVIS, and Visual Genome demonstrate its strong generalizability.

**Questions:**

Please to refer the weakness section:
1. Please demonstrate the significance of the difference between several modules of this work and other similar works
2. Please add a more detailed explanation about the memory module and ablation study about the stored proposal features


**Strengths And Weaknesses:**

Strength:
1. This work studies the problem of novel class discovery in the wild, which is insightful and valuable in terms of spurring future work.
2. The overall framework is intuitive and has good soundness, and extensive experiments demonstrate its effectiveness and good generalizability.

Weakness:
1. The idea of secondary classification head architecture is not novel. In [1] [2] they adopt a similar secondary classification head that parallels the known classification head and novel classification head. Can you clarify the difference between this work and theirs, and demonstrate the significance of the difference?
2. Table 3 shows the memory module contributes most to the performance, but the idea is still not very novel and similar to [3] [4], and the explanation about it (line 211 to line 218) and ablation experiment (line 308 to line 311) is too coarse, can you explain more? Line 215 says only the proposal feature from the last batch is stored in the memory, how about storing earlier proposal features?

[1] Fan Z, Ma Y, Li Z, et al. Generalized few-shot object detection without forgetting[C]//Proceedings of the IEEE/CVF Conference on Computer Vision and Pattern Recognition.

[2] Cao Y, Wang J, Jin Y, et al. Few-Shot Object Detection via Association and DIscrimination[J]. Advances in Neural Information Processing Systems, 2021

[3] Caron M, Misra I, Mairal J, et al. Unsupervised learning of visual features by contrasting cluster assignments[J]. Advances in Neural Information Processing Systems, 2020

[4] He K, Fan H, Wu Y, et al. Momentum contrast for unsupervised visual representation learning[C]//Proceedings of the IEEE/CVF conference on computer vision and pattern recognition

---

> ### Author Response · Authors · 2022-08-02
> **Response to Reviewer u1PQ**
>
> We are thrilled that the reviewer finds our work **insightful and valuable in terms of spurring future work**, experiments **extensive**, the overall framework **sound and intuitive**, as well as **effective and generalizable**. We are grateful for insightful comments; in the following, we address reviewer’s comments and questions.
>
> **The reviewer comments on the secondary classification head architecture and asks for clarification how it differs from prior work, in the review references as [1, 2].**
> While there are some similarities between the secondary classification heads proposed in [1, 2] to our work, we tackle a different problem and propose a rather different solution. Both [1, 2] tackle the task of  few-shot learning which is conceptually different to our task. Therefore, [1, 2] use explicit supervision in the form of labeled data for each “novel” class. In addition, [2] uses additional semantic supervision, based on WordNet lexical database to compute semantic similarity between novel and known classes. For both works, the set of novel classes is predetermined. This information is used to set the size of the secondary head. Ground truth pre-localized instances for each novel class are used to obtain a strong supervised training signal. This contrasts with our method that does not use any supervision for the novel classes and tackles a significantly more challenging problem of clustering class-agnostic object proposals under an unknown number of classes. We thus cannot set the secondary head size deterministically and have to focus on the unsupervised loss. Furthermore, we tackle the problem of discovering 1000+ novel (incl. fine-grained) classes with a long-tailed distribution, which is much more complex than in the aforementioned works that deal with 5 to 20 novel classes.
>
> [1] starts with a supervised pretraining of the R-CNN network, freezes it and attaches new RPN and RoI heads, including the secondary classification one. It then proceeds with a combination of fully-supervised loss and a KL regularizer that distills knowledge from the primary head to the secondary head for the known classes. [2] begins with a supervised pretraining followed by training a feature projector to align novel-object features. It then attaches a secondary classification head, freezes the feature extractors, and continues with a fully-supervised training with an auxiliary margin loss. By contrast, our method does not use any labeled supervison for novel classes. These challenges necessitated us to employ specific components that perform robustly in the absence of labels, such as constrained clustering with memory module, swapped assignments, downscaled supervised loss, constituting the core of our two-stage object detector.
> We have extended our related Work section with discussion of the few-shot learning and referenced [1, 2].
>
> **The reviewer (correctly) identifies our memory module as one of the key components needed for our method to work, asks for detailed explanation and refined ablation on storing features from earlier proposals (not only from the last batch).**
> Our method is indeed not the first to use a memory module to store features from the last batches (we make no such claims). With our ablation in Table 3, we aimed to demonstrate that the memory size is critical to the method's performance. As per the reviewer’s request, we revised the method section (subsection "Memory module") with a more detailed explanation of the memory module. We also added a refined memory size ablation to the supplementary section E and Table E1c. To answer the reviewer's question about the memory size employed for our experiments, in the memory module we store features of multiple past batches. More specifically, in all our experiments we use a first-in, first-out queue and store 100 last batches of the features (we mention this in Supplementary B, and we are grateful to the reviewer for pointing out this should be explained in the main text).

---

> > ### Author Response · Authors · 2022-08-09
> > **Further clarifications**
> >
> > Dear Reviewer, we hope our previous comment clarified your main concerns. We are open to further discussion for the remaining time.

---

> > ### Comment · Reviewer_u1PQ · 2022-08-09
> > **My Final Rating**
> >
> > The rebuttal solves most of my concerns. I will raise my rating to 5.

---

> > > ### Author Response · Authors · 2022-08-09
> > > **Thank you**
> > >
> > > We thank the reviewer for increasing their rating, and we are glad to have solved most of the reviewer's concerns.

---

### Official Review · Reviewer_yVTK · 2022-07-13

**Rating:** 7
**Confidence:** 4
**Soundness:** 3 good
**Presentation:** 3 good
**Contribution:** 3 good

**Summary:**

This paper tackles the problem of novel class discovery and detection. The authors propose a two-stage object detection network called Region-based NCDL. The proposed approach uses region proposals and is trained to classify them as belonging to either the seen classes or one of the unseen/novel classes.

Post-rebuttal - I have decided to update the rating to 7 after the authors' rebuttal to the initial reviews and updates to the paper.

**Questions:**

Please see the section above.

**Limitations:**

The authors have added a very brief discussion on the limitations of the proposed approach.

**Strengths And Weaknesses:**

The paper is mostly well written and the proposed approach achieves good results. I have the following questions/comments which the authors should address in the rebuttal and the updated submission.

1. In lines 114-115, the authors claim that in ZSL all the target classes are known and pre-determined. However, this is not true. Zero-shot learning methods do not need to assume knowledge of target classes apriori.

2. Something that's not clear to me currently is how exactly are the semantic class assignments made under clustering. Without using additional language information, how do the authors assign a semantic meaning to the discovered classes? The authors should provide the details of this in their rebuttal and add it to the final version.

3. In lines 185-187, the authors mention following [1] and [2] to generate cluster assignments. The authors should add some more details about these methods to make the paper more self-contained and easier to understand.

---

> ### Author Response · Authors · 2022-08-02
> **Response to Reviewer yVTK**
>
> We thank the reviewer for the overall positive review and insightful comments. We are excited that the reviewer finds our paper **well-written and results convincing**. In the following, we address the reviewer's questions and comments.
>
> **On reviewer’s comment that our statement on zero-shot learning is inaccurate.**
> We thank the reviewer for carefully reading our paper and spotting an inaccurate statement. We have updated the respective paragraph in the Related Work section.
>
> **Reviewer asks how exactly the semantic class assignments are made under clustering.**
> While the evaluation methodology was previously described in supplementary D, we in retrospect agree with the reviewer (as well as with reviewer B97E) that this should be explained in the main text for completeness. We have revised the main text (subsection "Inference and evaluation") to explain our evaluation setting. To match the predicted clusters with annotated classes we employ the Hungarian algorithm on top of predictions for localized ground truth object instances. We then discard the ground truth annotations, while keeping the resulting mapping. Finally, we perform inference to generate object proposals, classify them, and use the foregoing mapping to obtain classes that match the target dataset's classes and can be used for evaluation.
> We also highlight that we uploaded (anonymized) code for our method, experiments and evaluation: https://github.com/5da15f23f9/learning-to-discover-and-detect-objects.
>
> **The reviewer asks how cluster assignments (for pseudo-labeling) are generated, and asks for a revision of the method section in the paper.**
> As per reviewers' yVTK and B97E suggestions, we revised our Sec. 3 in the main text. We would refer the reviewer to the revised paper for (revised) detailed and self-contained explanation. For completeness, in the following we briefly summarize how clustering assignments are generated.
> We build on self-labeling via constrained clustering, where to each object proposal obtained from the RPN we assign a soft pseudo-labels (cluster assignments) q and use them to optimize for standard cross-entropy loss by training the network. To select pseudo-labels q, we employ a constrained clustering method, similar to [1, 2, 11] and generate such assignments that (i) their marginals closely match a pre-defined (lognormal) prior distribution, (ii) they minimize a clustering energy (cross-entropy loss) for the given class-probabilities p of proposals' features in the batch. This is posed as a constrained optimization problem and solved using the fast online Sinkhorn-Knopp algorithm [14]. Having obtained the pseudo-labels q, we then proceed to training the neural network using the cross-entropy loss.

---

> > ### Comment · Reviewer_yVTK · 2022-08-09
> > **Response to the rebuttal**
> >
> > I thank the reviewers for their response to the reviews. I believe that the rebuttal and the corresponding updates to the paper address my and other reviewers' questions well. In particular, the I appreciate that the authors have updated section 3 of the paper to add more details and make the inference procedure clearer. In light of this, I am increasing my initial rating.

---

> > > ### Author Response · Authors · 2022-08-09
> > > **Thank you!**
> > >
> > > We thank the reviewer for increasing their initial score, and we are happy that our rebuttal answered the reviewer's questions well.

---

### Author Response · Authors · 2022-08-02
**References**

The references in the rebuttal correspond to references in the updated manuscript. In some cases, the references have changed from the previous version, because of the added new work. As an example, SwAV has changed from [17] (previous version) to [11] (updated version).

---

### Meta-Review · Area_Chair_Nb9b · 2022-08-24

**Recommendation:** Accept
**Confidence:** Certain

**Metareview:**

The meta reviewer has carefully read the paper, reviews, rebuttals, and discussions. The meta reviewer appreciates the authors' efforts to respond and revise the paper. The authors did a good job of convincing the reviewers. The meta reviewer believes that open-world object detection using self-supervised learning is of interest to the community. And the authors clearly explain the motivation that object discovery is modeled as a clustering problem and solved by Sinkhorn clustering. The authors are suggested to polish the paper considering the reviewers' comments.

**Award:**

No

---

### Decision · Program_Chairs · 2022-09-14

Accept